# Altered Cortical Palmitoylation Induces Widespread Molecular Disturbances in Parkinson’s Disease

**DOI:** 10.3390/ijms232214018

**Published:** 2022-11-14

**Authors:** Juan F. Cervilla-Martínez, Juan J. Rodríguez-Gotor, Krzysztof J. Wypijewski, Ángela Fontán-Lozano, Tao Wang, Enrique Santamaría, William Fuller, Rebeca Mejías

**Affiliations:** 1Department of Physiology, School of Biology, University of Seville, Avenida de la Reina Mercedes, 6, 41012 Sevilla, Spain; 2Instituto de Neurociencias CSIC-UMH, Avenida Santiago Ramón y Cajal s/n, San Juan de Alicante, 03550 Alicante, Spain; 3Institute of Cardiovascular and Medical Sciences, University of Glasgow, Glasgow G12 8QQ, UK; 4School of Life Sciences, University of Dundee, Dundee DD2 5DA, UK; 5Instituto de Biomedicina de Sevilla, Campus Hospital Universitario Virgen del Rocío, Avda. Manuel Siurot, s/n, 41013 Sevilla, Spain; 6McKusick—Nathans Institute of Genetic Medicine and Department of Pediatrics, Johns Hopkins University School of Medicine, Baltimore, MD 21205, USA; 7Clinical Neuroproteomics Unit, Navarrabiomed, Hospital Universitario de Navarra (HUN), Universidad Pública de Navarra (UPNA), IDISNA, Irunlarrea Street, 3, 31008 Pamplona, Spain

**Keywords:** Parkinson´s disease, proteomics, cerebral cortex, palmitoylation, interactome

## Abstract

The relationship between Parkinson’s disease (PD), the second-most common neurodegenerative disease after Alzheimer’s disease, and palmitoylation, a post-translational lipid modification, is not well understood. In this study, to better understand the role of protein palmitoylation in PD and the pathways altered in this disease, we analyzed the differential palmitoyl proteome (palmitome) in the cerebral cortex of PD patients compared to controls (*n* = 4 per group). Data-mining of the cortical palmitome from PD patients and controls allowed us to: (i) detect a set of 150 proteins with altered palmitoylation in PD subjects in comparison with controls; (ii) describe the biological pathways and targets predicted to be altered by these palmitoylation changes; and (iii) depict the overlap between the differential palmitome identified in our study with protein interactomes of the PD-linked proteins α-synuclein, LRRK2, DJ-1, PINK1, GBA and UCHL1. In summary, we partially characterized the altered palmitome in the cortex of PD patients, which is predicted to impact cytoskeleton, mitochondrial and fibrinogen functions, as well as cell survival. Our study suggests that protein palmitoylation could have a role in the pathophysiology of PD, and that comprehensive palmitoyl-proteomics offers a powerful approach for elucidating novel cellular pathways modulated in this neurodegenerative disease.

## 1. Introduction

PD is the second-most common neurodegenerative disease. Its pathological hallmarks are the loss of dopaminergic neurons in the substantia nigra pars compacta (SNc) and the presence of Lewy bodies (LBs), insoluble intracytoplasmic neuronal inclusions containing misfolded α-synuclein. The loss of dopaminergic neurons in the SNc leads to a depletion of dopamine in the striatum, which in turn is responsible for the serious motor symptoms of PD [1]. As the disease evolves, neurodegeneration spreads to the rest of the central nervous system, including the cerebral cortex [2]. In addition to motor disturbances, non-motor symptoms that profoundly affect the quality of life of PD patients—such as REM sleep disturbances, mood, constipation, and loss of smell—can appear even before the development of motor symptoms [3]. Although the exact causes of PD are unknown, in 5–15% of cases, a mutation in a Parkinson´s disease gene (PARK gene) has been identified, and some environmental factors, such as certain pesticides, have also been linked to the risk of PD [1]. The characterization of the protein products encoded by PD-linked genes has provided important insights into the pathophysiology of familial and sporadic PD. Thus, alterations in mitochondrial function, vesicular transport, protein degradation and oxidative balance, among other cellular functions, have been postulated as pathways involved in the neurodegeneration associated with PD [4]. At present there is no cure for PD, and treatments simply aim to improve motor and non-motor symptoms [5].

S-acylation (called palmitoylation for simplicity through the manuscript) is a post-translational modification in which a palmitate molecule is reversibly conjugated to a cysteine residue of a protein via a thioester bond. Palmitoylation is regulated by enzymes which add palmitate (palmitoyl acyltransferases, PATs) or remove it (acyl protein thioesterases) from proteins [6]. Twenty-three PATs have been identified in mammals to date, but the knowledge about depalmitoylating enzymes remains limited [7,8]. Protein palmitoylation can regulate trafficking, stability and enzymatic activity of proteins, and dysregulation of this lipid modification has been implicated in several human diseases, including cancers, metabolic syndrome and infections [9,10]. More specifically, in the nervous system, several studies have linked alterations in palmitoylation to schizophrenia, Alzheimer´s disease (AD), intellectual disability, neuronal ceroid lipofuscinosis and Huntington’s disease [6,7,11,12,13,14,15,16]. It is well known that multiple neural signaling proteins, ion channels, cell adhesion molecules, SNARE proteins, vesicular trafficking proteins and G-protein receptors are palmitoylated [17,18]. Moreover, this lipid modification has been closely associated with neuronal development and synaptic plasticity [8,19,20]. However, a possible link between palmitoylation/depalmitoylation defects and PD has not been studied *in vivo*, and our limited understanding of this relationship comes from in-vitro and in-silico studies. Thus, palmitoylation of the dopamine transporter (DAT) and DJ-1, encoded by the PARK7 gene, have been described, and palmitoylation of LRRK2, encoded by the PARK8 gene, has been predicted [21,22,23]. In addition, it has recently been reported that ﻿upregulation of cellular palmitoylation alleviates the accumulation and neurotoxicity induced by α-synuclein [24].

In recent years, neuroproteomics has emerged as a powerful approach to characterize multiple proteotypes associated with neurodegeneration [25,26]. However, there is a lack of information about the impact of palmitoylation during the progression of the neurodegenerative process associated with PD. We therefore studied the cortical palmitoyl-proteome (palmitome) of PD and control subjects. After isolating palmitoylated proteins from cortical human samples using the Acyl-Rac assay, quantitative proteomics was performed to detect, identify and quantify palmitoylated proteins levels in both groups. In this pilot study, we characterized a set of 84 proteins that show altered levels of palmitoylation in the cerebral cortex of PD patients vs. controls. Interestingly, around 30% of these proteins were not previously known to be palmitoylated. These alterations in protein palmitoylation might have an impact on microtubule transport, inflammation, oxidative stress and tau pathology, among other physiological effects. Moreover, some of the proteins with altered palmitoylation levels in PD patients belong to the interactome of proteins encoded by most relevant PARK genes, suggesting that alterations in palmitoylation could contribute to pathological mechanisms causing PD. Using publicly available RNA-seq datasets, we show that the differentially palmitoylated proteins tend to map to different cell types in the cerebral cortex, but mainly in neurons. Our results demonstrate, for the first time in human brain tissue, a global deregulation of protein palmitoylation in the cortex of PD patients. We suggest that the deployment of brain palmitoyl-proteomics is an ideal approach that will support focused exploration of the imbalance in palmitoylation-dependent regulatory mechanisms associated with this disease.

## 2. Results and Discussion

### 2.1. Altered Palmitoylation Levels of Cortical Proteins in PD

To characterize the palmitoyl-proteome (palmitome) signature in the brain of PD patients, we first isolated palmitoylated proteins using the Acyl-RAC assay from cortical tissue derived from PD subjects and controls with no history of neurological disorders (Appendix A), and then performed label-free LC-MS/MS-based comparative proteomics [27,28]. Among 2284 putatively palmitoylated proteins identified in this proteomics study (Appendix A), 84 proteins had statistically significant differences in palmitoylation when comparing PD patients and control subjects (4% of the quantified palmitome; Log2(FC): ≥[0.6]; *p*-value: ≤0.05) (Appendix A). We will refer to these 84 proteins as differentially palmitoylated proteins (DPPs) henceforth. We compared our results (set of all palmitoylated proteins identified in human cortex and DPPs) with the database of S-palmitoylation SwissPalm [29]. SwissPalm contains information about the palmitoylation of all annotated proteins in *H. sapiens*, specifically whether a protein is known to be palmitoylated and the type of experiments that suggest the lipid modification (proteomics studies and/or further validation experiments). Approximately 20% of *Homo sapiens* proteins are palmitoylated, according to SwissPalm (Figure 1A). Focusing on the 2284 proteins identified in our cortical palmitome, more than 50% have previously been described to be palmitoylated according to SwissPalm (Figure 1B). Almost 70% of the DPPs are palmitoylated, according to the same database (Figure 1C). The increase in the percentage of palmitoylated proteins in the pie charts from Figure 1B,C compared to that of Figure 1A confirms the successful isolation of palmitoylated proteins before the proteomics analysis. In addition, these observations also suggest that our proteomics experiment identified novel palmitoylated proteins, although additional targeted studies are needed to validate this conclusion. In order to corroborate and broaden the findings of this study, future studies are needed to confirm our data in sex-balanced large cohorts and expand palmitome analysis across other brain areas relevant in the establishment and evolution of PD. In addition, comparison of palmitome data and whole proteome analysis will help to rule out differences in protein palmitoylation levels that are merely due to alterations in the total levels of proteins in PD patients. It is notable that global changes in protein palmitoylation were also found in patients with the neurodegenerative disorder neuronal ceroid lipofuscinosis [17].

From the 84 DPPs identified in this work, 33 proteins were more palmitoylated and 51 proteins were less palmitoylated in PD compared to the control group (Figure 2 and Appendix A).

In order to understand the biological and functional impact of the alterations in cortical palmitoylation levels in PD subjects, we performed GO (Gene Ontology) enrichment analysis [30] (summarized in Appendix A; adjusted *p*-value: ≤0.05). Figure 3A shows a heatplot with the 15 most relevant GO terms obtained in this analysis. In Figure 3B, DPPs (represented by their gene symbols) included within these GO terms are shown. Figure 3C displays a cnet plot to visualize the linkages of these DPPs with the most significantly altered biological functions in PD. In summary, key palmitoylated proteins are related to coagulation (*FGA*, *FGB* and *FGG*), dendritic spine, postsynapse and neuron projection organization (*ABHD17B*, *CAMK2B*, *CFL1*, *DBN1*, *DNM3*, *MAP6* and *YWHAH*) and cell-cell adhesion (*CSRP1* and *DSP*). Interestingly, ABDH17 proteins have depalmitoylating activity, but their palmitoylation has not been reported to date [31]. We observed an increase in the palmitoylation levels of the three polypeptides that constitute the fibrinogen molecule (encoded by genes *FGA*, *FGB* and *FGG*) in our PD sample cohort. Fibrinogen has essential roles in coagulation, inflammation, extracellular matrix physiology and tissue repair, having chaperone-like activity [32,33,34,35]. Fibrinogen is undetectable in the healthy brain, but it can deposit abundantly after blood-brain barrier (BBB) disruption in a variety of neurological diseases, suggesting a potential pathological role [36,37,38,39]. Fibrinogen deposition has not been explored in the brain of PD patients, although disruption of the BBB and inflammation may be involved in disease onset [40,41]. Alterations in normal clotting of blood have not previously been described in PD patients, although Sato et al. reported a possible relationship between the use of antiparkinsonian drugs and coagulation abnormalities [42]. Studies about fibrinogen levels in plasma and cerebrospinal fluid (CSF) in PD patients have led to conflicting results. Chen et al. reported no relationship between plasma levels of fibrinogen and the risk of developing PD [43], while other reports found high fibrinogen levels in plasma and CSF associated with PD [44,45,46]. Our data suggest that PD induces an up-regulation in the levels of palmitoylated fibrinogen polypeptides, at least at cortical level in the brain. Interestingly, palmitoylation of fibrinogen subunits has not been described to date, and consequences of this lipid modification in the functioning of the fibrinogen complex are unknown. Therefore, further experiments are needed to validate fibrinogen palmitoylation, and characterize its role in the onset and progression of PD.

To further study pathways affected by the alterations in the DPPs in PD patients, and complementing the results obtained by the GO enrichment analysis, we performed Metascape analysis (results summarized in Appendix A; *p*-values are provided in logarithmic based ten: “LogP”; *p*-value: ≤0.05). Metascape is a resource that incorporates a core set of default ontologies for enrichment analysis, including GO processes (GO), KEGG pathways (hsa), Reactome gene sets (R-HSA) and WikiPathways (WP) [47]. Interestingly, as shown in Figure 4, the Parkinson´s disease pathway (hsa05012 term) was the most affected route, according to KEGG Pathway Database. More specifically, nine of the DPPs are involved in PD-related molecular pathways, according to KEGG: *CAMK2B-*Calcium/Calmodulin Dependent Protein Kinase II Beta, *GPR37*-G Protein-Coupled Receptor 37, *MAPT*-Microtubule Associated Protein Tau, *NDUFV1*- NADH: Ubiquinone Oxidoreductase Core Subunit V1 (in mitochondrial Complex I), *TUBB2A*-Tubulin Beta 2A Class IIa, *TXN*-Thioredoxin, *UQCRC1*- Ubiquinol-Cytochrome C Reductase Core Protein 1 (in mitochondrial complex III), *TUBB3*- Tubulin Beta 3 Class III, and *TUBB4A*-Tubulin Beta 4A Class IVa (Gene Symbols- Protein coding). A visualization of the PD pathway can be seen at https://www.kegg.jp/kegg-bin/show_pathway?hsa05012/816/2861/4137/4723/7280/7295/7384/10381/10382 (accessed on 9 October 2022) with altered DPPs proteins (in a red rectangle and in red font color) and their interactions with proteins encoded by PD-associated genes DJ-1, PaelR, Parkin, PINK1, SNCA and UCHL1 (in a black rectangle and in red font color). It is notable that dysfunction of mitochondrial complex I has been recently linked to the loss of dopaminergic neurons and metabolic changes in the substantia nigra of mice, causing a human-like parkinsonism [48], and that low activity of mitochondrial complexes I, II and III has been reported in lymphocytes and platelets of PD patients [49,50]. Thioredoxin has antioxidant activity that protects dopaminergic neurons in animal models of PD [51], and its palmitoylation has been shown to modulate its activity [52,53]. Although the role of Tubulin Beta in PD is not well defined, its degradation seems to be regulated by Parkin [54], and its palmitoylation has been suggested [55]. Metascape analysis also predicted changes in MAPK signaling for integrins (R-HSA-354194; *FGA*, *FGB*, *FGG* and *RAP1B*) and cellular responses to stimuli (R-HSA-8953897; *ATP6V1A*, *CAMK2B*, *CSRP1*, *EIF2S3*, *HSPA2*, *RPL7A*, *TKT*, *TUBB2A*, *TXN*, *KHSRP*, *TUBB3*, *TUBB4A* and *PRDX3*), among other pathways. Taken together, the analyses of our palmitome study in PD cortex highlights both well-known and novel molecular pathways that could be affected in this disease.

### 2.2. Network-Driven Palmitoyl-Proteomics Reveals an Imbalance in Cytoskeletal Architecture and Mitochondrial Homeostasis in PD

To enhance the biological outcome of our palmitome approach, proteome-scale interaction networks were constructed using STRING (*p* ≤ 0.05) [56] and QIAGEN Ingenuity Pathway Analysis (IPA). The outcome obtained by the STRING analysis of DPPs can be seen at https://version-11-5.string-db.org/cgi/network?taskId=bx7tBdv4gtLt&sessionId=bAXrCBiw5ZOI (accessed on 9 October 2022). Figure 5A shows a functional and physical protein-protein interaction (PPI) network of DPPs, and Figure 5B displays just the physical PPI network of DPPs obtained using the STRING tool. In this analysis, one of the affected molecular functions involved constituents of the cytoskeleton. Tubulin beta polypeptides (encoded by *TUBB2A*, *TUBB3* and *TUBB4A*), Microtubule Associated Protein Tau (encoded by *MAPT*) and Calcium/Calmodulin Dependent Protein Kinase II Beta (encoded by *CAMK2B*) are found in the same PPI network, with lower levels of palmitoylation in PD patients compared to the controls (Figure 5A). Tubulin alpha and beta heterodimers are structural components of the microtubules and have key roles on multiple cellular functions, such as the intracellular molecular trafficking [57]. Microtubule Associated Protein Tau, usually named Tau for simplicity, is responsible for stabilizing the microtubules, and its pathological deposition is known as tauopathy. In tauopathies, the soluble tau protein separates from microtubules, and creates abnormal, fibrillar structures of aggregated, hyperphosphorylated and acetylated tau [58,59]. Some tauopathies are characterized by parkinsonism, but PD was not initially considered to be a typical tauopathy. However, recent studies have demonstrated increasing evidence of tau pathology in PD. Interestingly, hyperphosphorylated tau proteins have been shown to interact with α-synuclein promoting aggregation and fibrilization to each other, accelerating the formation of Lewy bodies and axonal transport dysfunction [60,61,62,63]. Our study supports the existence of tau pathology in the cortex of PD patients, and suggests that tau could be palmitoylated in vivo. Calcium/Calmodulin Dependent Protein Kinase II Beta (CAMKIIβ), whose possible palmitoylation has not been studied, can associate with the actin cytoskeleton (microfilaments), and has a role in regulating dendritic arborization and synapse density [64,65]. Interestingly, cytoskeleton dysfunctions in PD have been suggested [66,67,68], and it would be valuable to unravel the consequences of altered palmitoylation on the function of this crucial cellular structure. 

In accordance with the data obtained from Metascape (MAPK signaling for integrins, R-HSA-354194; Figure 4 and Appendix A), STRING analysis also predicted a functional and physical PPI network, including the fibrinogen components and Ras-Related Protein Rap-1b (encoded by *RAP1B* gene) (Figure 5A,B). Modulation of the binding of fibrinogen to integrins by Rap-1b has been studied in platelet aggregation and megakaryocytes [69,70], but no information is available related to the interactions of these proteins in the brain.

To further study the functional connection of our proteomics data with signaling hubs, we used IPA. To enrich our proteomic dataset, we included the qualitative palmitoylated proteins (QPPs, 66 proteins listed in Appendix A) to the 84 DPPs in this analysis (total input dataset of 150 proteins). QPPs are proteins whose palmitoylation is almost undetectable in the cortex of one of the groups, either the control or the PD group, but detectable in the other one, and therefore the differences in palmitoylation levels are quite striking, but cannot be quantified. As shown in Figure 6, protein network analysis revealed that the deregulated cortical palmitome of PD patients is functionally associated to survival mediators such as Protein Kinase B (Akt) and Protein Phosphatase 2 (PP2A). Both Akt and PP2 have been implicated in a variety of neurodegenerative disorders, including PD. Specifically, Akt has been reported to promote neuronal survival [71] and PP2A to modulate mitochondrial function [72,73]. Although palmitome changes were predicted to inhibit AKT (Figure 6), additional research is needed to characterize the potential crosstalk between cellular palmitoylation and survival mediators in the context of PD. 

Our functional interactome data also recognized a protein set and their PPI interactions that affect mitochondrial homeostasis, specifically the mitochondrial complex I functionality in the cortex of PD subjects (Figure 7). In accordance with this information, Metascape analyses also identified a mitochondrial disruption linked to dyshomeostatic palmitoylation (Parkinson´s disease pathway, hsa05012 term, Ubiquinone Oxidoreductase Core Subunit V1 and Ubiquinol-Cytochrome C Reductase Core Protein 1). Peroxiredoxin 3 (PRDX3) is associated with the mitochondria, where it exerts its antioxidant role [74]. Both altered mitochondrial complexes activity and oxidative stress have been described as molecular mechanisms underlying the pathophysiology of PD [48,49,50,75]. On the other hand, O-Linked N-Acetylglucosamine (GlcNAc) Transferase (OGT), whose palmitoylation levels are decreased in the cortex of PD patients (see Figure 7), catalyzes the post-translational modification of proteins named O-GlcNAcylation. O-GlcNAcylation modulates protein function; for instance, the ability of the palmitoyl acyltransferase zDHHC5 to recruit a substrate [76]. Inhibition of OGT might be neuroprotective against AD, and tau protein can be O-GlcNAcylated, at least in animal models [77,78,79]. GlcNAcylation on α-synuclein has also been established, which has an inhibitory effect on fibril formation, neurotoxicity, and aggregation [80,81]. Although there is no information about palmitoylation of OGT in the literature, palmitoylation of PRDX3 has been described in mice [82]. According to IPA knowledgebase, PRDX3 and OGT are functionally linked to NF-κβ, a key regulator of inflammation and apoptosis, whose dysregulations could be involved in the onset of PD [83,84]. Thus, NF-κβ activation in the brain of PD patients could contribute to chronic neuroinflammation, and its inhibition has been proposed to slow down the progression of this disease [85,86]. Accordingly, IPA predicts a potential activation of NF-κβ in the cortex of PD patients. NF-κβ is not known to be palmitoylated. In summary, network data from STRING and IPA analyses indicate that cytoskeletal-mitochondrial axis could be modulated, at least in part, by levels of protein palmitoylation in the cortex of PD patients. Some studies have pointed to a relationship between cytoskeleton alterations and mitochondrial dysfunction in PD [87,88].

### 2.3. Differential Palmitoyl-Proteins Are Components of Interactomes Associated to PD Genes and Are Located in Multiple Cell Types 

Discovery of unexpected relationships between apparently unrelated and PD-linked proteins may be a powerful strategy for the characterization of novel PD targets and pathways. We explored whether the interactome of well-established PD-related proteins were indeed potentially interconnected with the DPPs detected at the cortical level. As shown in Figure 8 and Appendix A, we identified several DPPs that are known interactors of proteins encoded by PD-linked genes (SNCA, α-synuclein; LRRK2, leucine-rich repeat kinase 2; GBA, glucocerebrosidase; UCHL1, ubiquitin C-terminal hydrolase L1; PINK1, PTEN-induced kinase 1). Strikingly, we found that PRDX3 (Peroxiredoxin 3) is a common interactor of α-synuclein, LRRK2 and DJ-1 (Figure 8A and Appendix A). As discussed earlier, PRDX3 has an antioxidant role in the mitochondria, and it is known to be palmitoylated in mice [74,82]. PRDX3 is phosphorylated by LRRK2 [89], and its overexpression reduces the neurotoxicity associated to LRRK2 kinase mutant G2019S in flies [90]. We highlight that it will be of interest to study whether PRDX3 is palmitoylated in human samples, and the influence of this lipid modification on the antioxidant activity of this enzyme, since PDRX5 palmitoylation is established to reduce its antioxidant capacity in mitochondria [84]. 

Secondly, we found that there are 17 DPPs (listed in Appendix A) which are common interactors of GBA, UCHL1 and PINK1 (Figure 8B). Remarkably, three DPPs—CFL1 (cofilin 1), TKT (transketolase) and DBN1 (Drebrin 1)—are shared interactors of most of the PD-linked proteins studied (LRRK2, DJ-1, PINK1, GBA and UCHL1) (Figure 8C and Appendix A). Cofilin 1 is an actin-binding protein that can regulate mitochondrial function [91]. Drebrin 1 is also a cytoplasmic actin-binding protein [92], and transketolase is a key enzyme in the non-oxidative branch of the pentose phosphate pathway [93]. However, the precise mechanism by which these enzymes are involved in PD is unknown. 

Lastly, to explore whether alterations in protein palmitoylation occur in specific cell types in the cortex of PD patients, we compared our differential dataset with public available cell-type RNA-seq and proteomics datasets [94,95]. As shown in Appendix A and Appendix A, a high proportion of DPPs and QPPs tend to be widely distributed across multiple cell-types, with neuronal proteins being the most enriched protein set.

## 3. Materials and Methods

### 3.1. Subjects and Samples

Post-mortem cerebral cortex samples were obtained from the Banner Sun Health Research Institute Brain and Body Donation Program of Sun City, Arizona (https://www.bannerhealth.com/es/services/research/locations/sun-health-institute/programs/body-donation (accessed on 9 October 2022)) after approval by the Banner Sun Health Research Institute Tissue Allocation Committee (IRB approval from the wcg IRB of Puyallup, Washington; study number 1132516). Controls were defined as a subject without dementia or parkinsonism during life, and without a major neuropathological diagnosis. A Parkinson’s disease (PD) patient was defined as having two of the three cardinal clinical signs of resting tremor, muscular rigidity and bradykinesia, along with pigmented neuron loss and Lewy bodies in the substantia nigra. We included in this study four PD and four control cortical samples. There were no differences between control and PD groups regarding postmortem interval to obtain biopsies (around 3 h after death), senile plaque density or neurofibrillary tangle density. Additional information on subject characteristics is available on Appendix A.

### 3.2. Acyl-RAC

The isolation of palmitoylated proteins was based on the Acyl-Rac method [96,97]. Samples were homogenized in blocking buffer (100 mM HEPES, 1.0 mM EDTA, 2.5% SDS, 1% MMTS, pH 7.5) and incubated for 4 h at 40 °C and with agitation at 1200 rpm. Samples were precipitated in acetone to eliminate the excess of MMTS, and then pellets were washed with acetone 70%. Pellets were dried and resolubilized in binding buffer (100 mM HEPES, 1.0 mM EDTA, 1% SDS, pH 7.5) at 40 °C and 1400 rpm for 2 h. Palmitoylated proteins were captured on thiopropyl sepharose beads (T8387, Sigma Aldrich) in the presence of 200 mM hydroxylamine, pH 7.4, by rotating for 2.5 h at room temperature. The beads were then washed five times in binding buffer, and the proteins were eluted in SDS PAGE loading buffer with 100 mM DTT.

### 3.3. Label Free Proteomics

To perform label-free LC-MS/MS-based comparative proteomics, proteins isolated from Acyl-Rac technique had brief electrophoresis, followed by in-gel digest and reductive alkylation. Trypsin-digested peptides were processed for mass spectrometry using a Thermo Scientific LTQ Velos Pro Orbitrap, following methodology described before at the FingerPrints Proteomics Service at the University of Dundee (https://proteomics.lifesci.dundee.ac.uk/ (accessed on 9 October 2022)) [27]. The raw MS/MS spectra searches were processed using the MaxQuant software and default settings (v 1.6.7.0) [98] and searched against the Uniprot proteome reference for Homo sapiens (Proteome ID: UP000005640_9606). 

### 3.4. Data Preprocessing

Appendix A contains the quantification and identification of proteins from the proteomics experiment. To obtain the list of differential palmitoylated proteins (DPPs), analysis of the LFQ (label-free quantitation) intensities proteins were performed using the software “R” and the integrated development environment “RStudio” [99], through packages found in the repositories CRAN (“The Comprehensive R Archive Network”) [100] and Bioconductor [101]. The packages “tidyverse” [102], “cowplot” [103], “gridExtra” [104] and “ggpubr” [105] were used for data transformation and plotting. “Readxl” package [106] was used for opening Excel files in RStudio IDE. Value imputation was performed as follows: data was imputed in a way that those proteins that did not have at least two values for each group (50%) were removed. The average expression value per group was calculated for each protein, filling missing values and completing the data matrix. After imputation, the “NormalyszerDE” package [107] was used to evaluate different data normalization approaches. The dataset with log2(FC) transformation and quantile normalization was selected. The selection of this type of normalization was corroborated by other statistical metrics such as Pearson and Spearman correlation, Q-Q plots, MA plots, etc., generated by the statistical summary of the package. Looking at the LFQ intensities, certain proteins had an interesting pattern of presence/absence of signal in controls and PD samples. These proteins, named through the manuscript “qualitative palmitoylated proteins” (QPPs), showed values in at least three samples in one group (either control or PD), and missing values (value of 0) in at least three samples in the other group. Since these proteins had insufficient values for quantitation in both conditions, they could not be used for the study of DPPs. Nevertheless, we considered that these candidates have potential biological interest, as the presence or absence of these proteins suggest drastic changes in the levels of palmitoylation between both experimental groups. 

### 3.5. Data Analysis

The “limma” package [108] was applied to obtain the DPPs. Proteins with a Log2(FC): ≥[0.6] and a *p*-value: ≤0.05 were selected. A heatmap of differential palmitoylated proteins was plotted using the “pheatmap” package [109]. Z-score normalization [110] was calculated to rearrange DPPs used in the heatmap. The SwissPalm database gathers information about the palmitoylation of proteins in the UniprotKB database [29]. We used this database to examine the previous knowledge about palmitoylation of the proteins identified in our cortical palmitome study. 

For functional enrichment analysis, the “clusterProfiler” package [111,112] and “org.Hs.eg.db” annotation package [113] were used. The “enrichGO” function from “cluster Profiler” was applied to search terms from Gene Ontology (GO) database [30,114] using DPPs. In this case, the method for determining the adjusted *p*-value was the false discovery rate (FDR) [115], with a threshold of 0.05. To further characterize the alteration in metabolic pathways, the Metascape web tool was used [47] to simultaneously query the KEGG databases [116,117,118], a set of databases that gather information about diverse species, as well as mapping tools for network visualization and molecular pathways; Reactome [119], a database which provides curated information about a wide range of biological processes in *Homo sapiens*; and WikiPathways [120], a database that includes molecular pathways linked to rare diseases and lipid metabolism, among other biological pathways. For this purpose, a list of DPPs translated into gene symbols was generated using the web tool BioDBnet [121]. Ten of the DPPs and 9 of the QPPs (see Appendix A) could not be translated into gene symbols, probably due to poor annotation. Within Metascape options, the customized analysis was selected, where only the databases previously mentioned and a *p*-value threshold of 0.05 were selected; all other enrichment options were set as default. The study of protein-protein interactions (PPI) within DPPs was performed using the STRING database and software [56]. To obtain the network, the list of DPPs, applying a *p*-value of 0.05 as a threshold, was used in combination with the color and intensity of the aureoles based on log2(FC), depending on their up (red) or down (green) palmitoylated levels. To complement the functional outputs, the Ingenuity Pathway Analysis software (IPA) from QIAGEN was applied [122], exclusively using the database information of experimental and predictive origin regarding central nervous system in order to be confident about the potential affected signaling routes. 

To interlock our DPPs dataset with interactomes from relevant PD-associated genes, experimentally demonstrated interactors of these PARK genes were downloaded from the BIOGRID database [123]. The packages “Venndetail” [124] and “UpSetR” [125] were used to visualize the interactomes through Venn diagrams and upset plots. To map DPPs signature in specific brain cell-types, public RNA-seq and proteomics datasets were used [94,95].

## 4. Conclusions

To the best of our knowledge, these results demonstrate, for the first time, alterations in protein palmitoylation levels in human PD tissue, suggesting a possible role of this posttranslational modification in the pathophysiology of PD. Future palmitoyl-proteomic approaches and palmitoylation assays in human samples could generate new knowledge to better understand the involvement of palmitoylation in PD. Our results confirm the role of well-stablished molecular pathways previously linked to PD, such as mitochondrial dysfunction, oxidative stress and inflammation. In addition, our proteomics approach pointed to fibrinogen and cytoskeleton proteins as novel targets linked to neurodegeneration in PD.

## Figures and Tables

**Figure 1 ijms-23-14018-f001:**
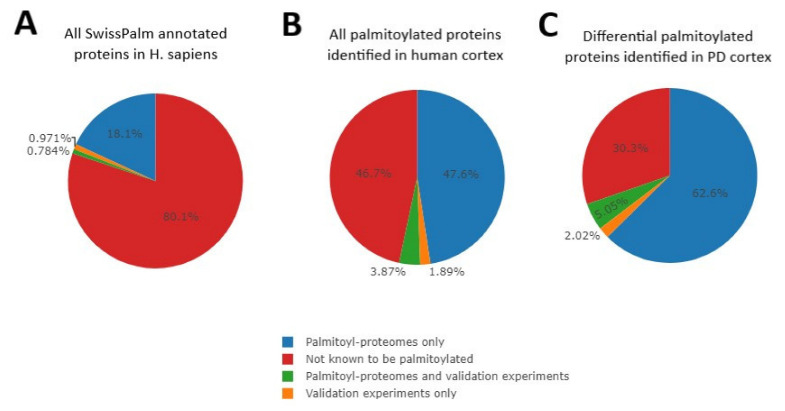
Pie charts representing the percentage of proteins per category (palmitoylated with different levels of validation, in blue, green or orange, or non-known to be palmitoylated, in red) included in the SwissPalm database for *Homo sapiens* (**A**), for all 2284 palmitoylated proteins detected in our workflow study (**B**), and for the 84 differentially palmitoylated proteins in PD patients identified (**C**).

**Figure 2 ijms-23-14018-f002:**
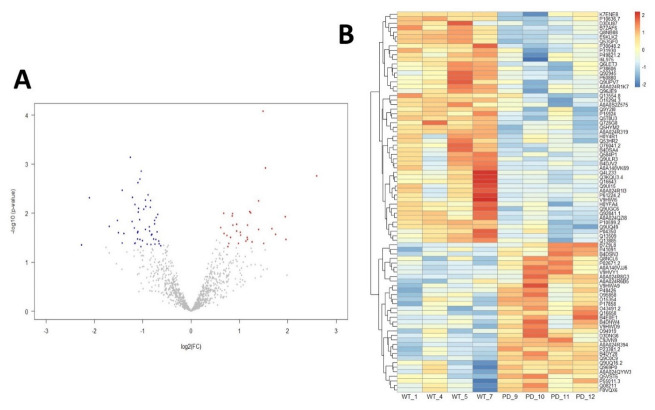
(**A**) Volcano plot showing proteins that are more palmitoylated (red) and less palmitoylated (blue) in cerebral cortex from PD patients. Proteins with no significant changes among groups are represented in grey; (**B**) Heatmap showing differential palmitoylated proteins arranged by Z-score. The UniProt accession number for each protein is shown.

**Figure 3 ijms-23-14018-f003:**
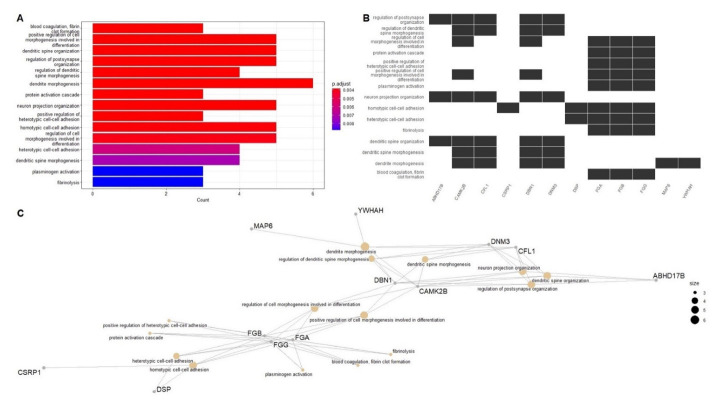
(**A**) Heatplot showing the 15 most significant terms obtained by performing GO enrichment analysis. (**B**) Barplot showing the DPPs (represented by their gene symbols) and their relationship with the 15 most relevant GO terms. (**C**) Cnet plot of significantly enriched GO terms.

**Figure 4 ijms-23-14018-f004:**
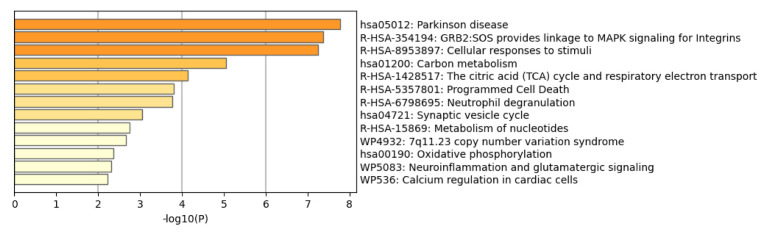
Barplot from Metascape analysis showing the 13 most relevant and non-redundant biofunctions affected by the palmitoylation changes in the DPPs detected at cortical level in PD.

**Figure 5 ijms-23-14018-f005:**
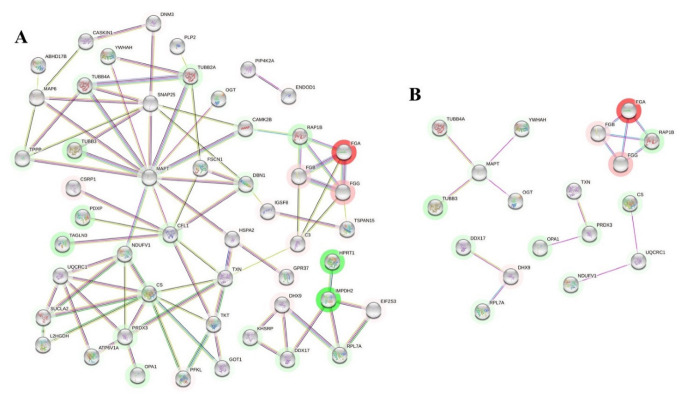
(**A**) Functional and physical protein-protein interaction (PPI) network of DPPs (represented by their gene symbols) obtained with STRING analysis. (**B**) Only physical PPI network of DPPs (represented by their gene symbols) obtained with the STRING tool are shown. Green and red halos represent down- or up-regulation of the palmitoylation levels, respectively. Color intensity of the halo is in accordance with the fold-change.

**Figure 6 ijms-23-14018-f006:**
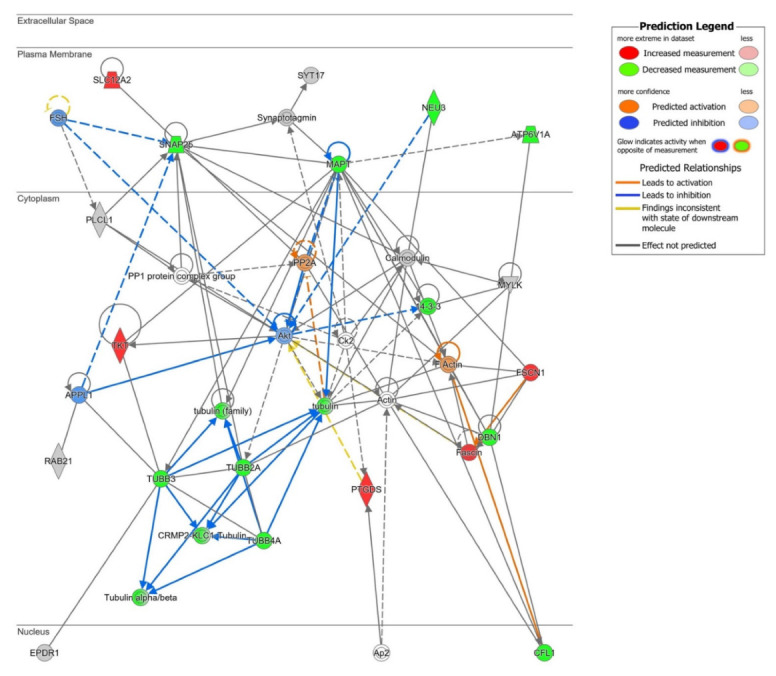
High-scoring protein interactome map focused on cell survival (Score 38). Visual representation of the relationships between DPPs and QPPs with functional interactors. DPPs are highlighted in red when palmitoylation is up-regulated, and in green when palmitoylation is down-regulated in PD cortex. Dataset molecules that did not pass the analyses cutoffs are shown with a grey background. Continuous and discontinuous lines represent direct and indirect interactions, respectively. Predicted interactors of DPPs and QPPs, not part of the dataset, are shown in white. See complete QIAGEN IPA legend at https://qiagen.my.salesforce-sites.com/KnowledgeBase/articles/Knowledge/Legend (accessed on 9 October 2022).

**Figure 7 ijms-23-14018-f007:**
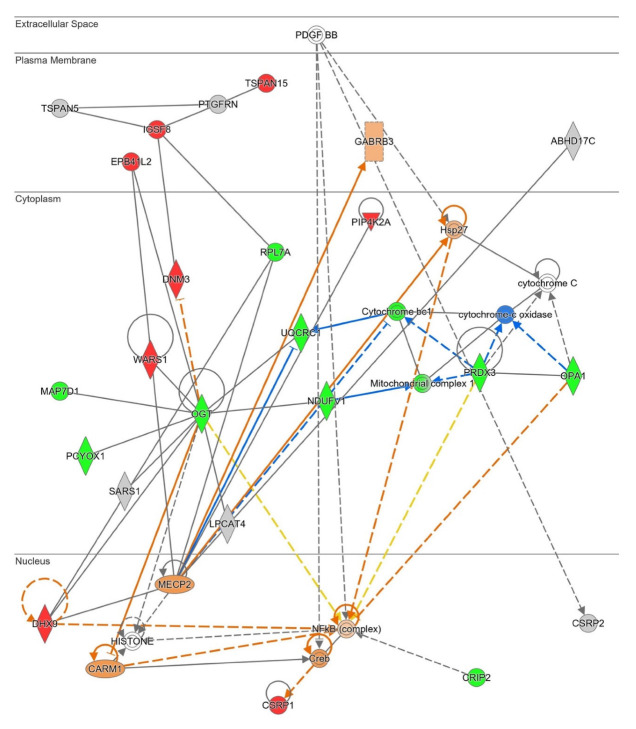
High-scoring protein interactome map unveiling a mitochondrial imbalance (Score 57). Visual representation of the relationships between DPPs and QPPs with functional interactors. DPPs are highlighted in red when palmitoylation is up-regulated, and in green when palmitoylation is down-regulated in the cortex of PD patients. Dataset molecules that did not pass the analyses cutoffs are shown with a grey background. Continuous and discontinuous lines represent direct and indirect interactions, respectively. Predicted interactors of DPPs and QPPs, not part of the dataset, are shown in white. See complete QIAGEN IPA legend at https://qiagen.my.salesforce-sites.com/KnowledgeBase/articles/Knowledge/Legend (accessed on 9 October 2022).

**Figure 8 ijms-23-14018-f008:**
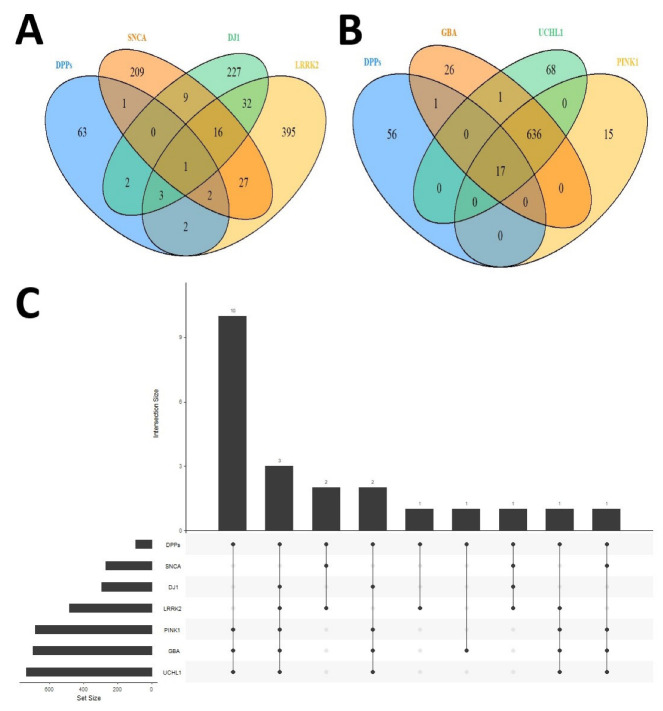
(**A**) Venn diagram showing the overlap between SNCA (α-synuclein), LRRK2 and DJ-1 interactomes with DPPs. (**B**) Venn diagram showing the overlap between GBA, UCHL1 and PINK1 interactomes with DPPs. (**C**) Upset plot showing the sizes of each interactome, and number and overlaps between SNCA, DJ1, LRRK2, PINK1, GBA and UCHL1 interactomes and DPPs.

## Data Availability

MS data and search results files were deposited in the Proteome Xchange Consortium via the JPOST partner repository [126] with the identifier PXD037352 for ProteomeXchange and JPST001888 for jPOST (for reviewers: https://repository.jpostdb.org/preview/14537275896347bdb27be4f (accessed on 9 October 2022); Access key: 2985).

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
