# Peer review of "Altered Cortical Palmitoylation Induces Widespread Molecular Disturbances in Parkinson’s Disease"

_ijms, 2022, doi:10.3390/ijms232214018_

Round 1

Reviewer 1 Report

In this manuscript, Cervilla-Martinez and colleagues have analysed the palmitoylation-induced alterations on proteins form Parkinson’s disease cortex. The study presents dysregulations in palmitoylation of certain PD associated proteins along with other non-related proteins identified by mass spectrometry analysis. The presented results are interesting, however the study presents some limitations which should be discussed.

Major concerns:

-The analysis is done only on 4 samples with a restriction of brain region-cortex and other regions of the brain have not been analysed, therefore the prognostic potential is limited.

-Palmitoylated proteins abundance by MS should be compared with whole proteome analysis and normalization is preferable.

-Label-free quantification offers a semi-quantitative analysis, for quantitation other methods of labelling can be easily applied and therefore increase the accuracy of the analysis

Minor concerns:

-proof reading is required, some letters are missing throughout the text and some phrases need some alterations

-figure resolution is quite low in the present file, for final processing this is an important aspect

Author Response

Major concerns:

- The analysis is done only on 4 samples with a restriction of brain region-cortex and other regions of the brain have not been analysed, therefore the prognostic potential is limited.

-Palmitoylated proteins abundance by MS should be compared with whole proteome analysis and normalization is preferable.

We agree with the reviewer about the points raised in his/her comments.

We are aware that a similar study should be done in other brain regions in the future. According to this idea, we had the following comment in line 133 in the original manuscript: “It would be interesting to compare the palmitoylation profile across different brain regions in PD patients in the future to provide a better understanding of the role of this post-translational modification in the origin and progression of this disease as well as its correlation with the neuropathological grading.”

We also agree with the reviewer that the number of samples used in this study is only 4 per group. The principal reason for this limitation is that funding for the project was limited, and the acquisition of human samples and proteomics studies are expensive. We plan to increase the number of samples in our future studies. The scope of this study was to explore whether PD patients showed alterations in palmitoylation, and further studies are now undergoing to expand the type and number of samples. However, despite the small number of samples, we have only highlighted in the manuscript proteins that showed statistical significance between groups (p-value: ≤0.05) and with a Log2(FC): ≥[0.6]. Besides, there are publications on neuroproteomics in humans using a limited number of samples:

  • Ayyadevara S et al. Proteins that mediate protein aggregation and cytotoxicity distinguish Alzheimer's hippocampus from normal controls. Aging Cell (2016), 15:924-39.
  • Basso M et al. Proteome analysis of human substantia nigra in Parkinson's disease. Proteomics (2004), 4:3943-52.
  • Begcevic I et al. Semiquantitative proteomic analysis of human hippocampal tissues from Alzheimer's disease and age-matched control brains. Clin Proteomics (2013), 10:5.
  • Jin J et al. Proteomic identification of a stress protein, mortalin/mthsp70/GRP75: relevance to Parkinson disease. Mol Cell Proteomics (2006), 5:1193-204.
  • Klettner A et al. Reduction of GAPDH in lenses of Parkinson's disease patients: A possible new biomarker. Mov Disord (2017), 32:459-462.
  • Shi Q et al. Proteomics analyses for the global proteins in the brain tissues of different human prion diseases. Mol Cell Proteomics (2015), 14:854-69.
  • Werner CJ et al. Proteome analysis of human substantia nigra in Parkinson's disease. Proteome Sci (2008), 6:8.

We again agree with the reviewer that the comparison of palmitomes with whole proteome analysis is a more complete approach for our studies. Again, we could not perform whole proteome analyses due to the limitation in our funding, but these studies will be included in future experiments to i) check steady-state protein levels and ii) compare differential palmitomes across different brain structures related to PD evolution. It is worth pointing out that although total protein levels were not quantified, our study has revealed proteins that could be palmitoylated that have not been described before as palmitoylation substrates. Besides, our studies revealed not only global changes in palmitoylation levels of proteins but also changes in palmitoylation of proteins that are experimentally demonstrated as physical interactors of PD-related proteins for the first time.

We have modified the sentence mentioned above (in line 133) according to the reviewer's comments: “In order to corroborate and broaden the findings of this study, future studies are needed to confirm our data in sex-balanced large cohorts and expand palmitome analysis across other brain areas relevant in the establishment and evolution of PD. In addition, com-parison of palmitome data and whole proteome analysis will help to rule out differences in protein palmitoylation levels that are merely due to alterations in total levels of the proteins in PD patients.”

-Label-free quantification offers a semi-quantitative analysis, for quantitation other methods of labeling can be easily applied and therefore increase the accuracy of the analysis.

As mentioned before the scope of our study was to test the hypothesis that protein palmitoylation could be altered in PD and further research will be performed to deep and validate the results shown in this manuscript. We agree with the reviewer´s comment about the deep proteomic profiles obtained with isobaric approaches. Of course, future studies will include iTRAQ or TMT labeling techniques, but also it would be necessary to evaluate the possibility to apply recent data-independent acquisition workflows in the context of palmitoylation, that allow the inclusion of a higher number of samples avoiding the use of a pooling strategy.

In any case, the FingerPrints Proteomics Service at the University of Dundee (where our Proteomics experiment was carried out) and other Proteomic Units, have successfully applied label-free quantitation, as referenced in the manuscript:

- Aghamaleky Sarvestany, A et al. Label-Free Quantitative Proteomic Profiling Identifies Disruption of Ubiquitin Homeostasis as a Key Driver of Schwann Cell Defects in Spinal Muscular Atrophy. J. Proteome Res. 2014, 13, 4546–4557,

- Cox, J et al. Accurate Proteome-Wide Label-Free Quantification by Delayed Normalization and Maximal Peptide Ratio Extraction, Termed MaxLFQ. Mol. Cell. Proteomics 2014, 13, 2513–2526, doi:10.1074/MCP.M113.031591.

 Minor concerns:

-proof reading is required, some letters are missing throughout the text and some phrases need some alterations.

We thank the reviewer for the comment. The manuscript has been reviewed and changes are highlighted in the text.

-figure resolution is quite low in the present file, for final processing this is an important aspect.

We thank the reviewer for the advice. Original figures with a higher resolution will be provided to the Journal to improve the quality of the figures in the final version.

Reviewer 2 Report

I read with very interest the manuscript (ID: ijms-1997654) entitled "Altered cortical palmitoylation induces widespread molecular 2 disturbances in Parkinson´s disease".

In this work, the Authors demonstrated alterations in protein palmitoylation levels in human Parkinson´s disease cerebral cortex, suggesting a possible role of this posttranslational modification in PD onset and progression.

The Abstract, Methods and Results accurately described the content of the article. The Results were properly discussed

In my opinion, this paper is acceptable in IJMS journal with minor issues, as it follows:

-          Line 5. PARK gene: The Authors must define this acronym

-          Line 91. PD patients vs. controls. I think the point after vs should be removed

-          Line 189. In Figure 3 legend, (A) must be replaced with (C)

Author Response

Comments and Suggestions for Authors

I read with very interest the manuscript (ID: ijms-1997654) entitled "Altered cortical palmitoylation induces widespread molecular 2 disturbances in Parkinson´s disease".

In this work, the Authors demonstrated alterations in protein palmitoylation levels in human Parkinson´s disease cerebral cortex, suggesting a possible role of this posttranslational modification in PD onset and progression.

The Abstract, Methods and Results accurately described the content of the article. The Results were properly discussed.

We appreciate the comments of the reviewer and have made the changes suggested in the revised version of the manuscript.

In my opinion, this paper is acceptable in IJMS journal with minor issues, as it follows:

-          Line 5. PARK gene: The Authors must define this acronym

-          Line 91. PD patients vs. controls. I think the point after vs should be removed

-          Line 189. In Figure 3 legend, (A) must be replaced with (C)
